# Towards objectively quantifying sensory hypersensitivity: a pilot study of the "Ariana effect"

Vassilis N. Panagopoulos[1], Deanna J. Greene[2], Meghan C. Campbell[3] and Kevin J. Black[4]

[1] Department of Psychiatry, Washington University School of Medicine, St. Louis, MO, USA
[2] Departments of Psychiatry and Radiology, Washington University School of Medicine, St. Louis, MO, USA
[3] Departments of Neurology and Radiology, Washington University School of Medicine, St. Louis, MO, USA
[4] Departments of Psychiatry, Neurology, Radiology, and Anatomy & Neurobiology, Washington University School of Medicine, St. Louis, MO, USA

## ABSTRACT

**Background.** Normally one habituates rapidly to steady, faint sensations. People with sensory hypersensitivity (SH), by contrast, continue to attend to such stimuli and find them noxious. SH is common in Tourette syndrome (TS) and autism, and methods to quantify SH may lead to better understanding of these disorders. In an attempt to objectively quantify SH severity, the authors tested whether a choice reaction time (CRT) task was a sensitive enough measure to detect significant distraction from a steady tactile stimulus, and to detect significantly greater distraction in subjects with more severe SH.

**Methods.** Nineteen ambulatory adult volunteers with varying scores on the Adult Sensory Questionnaire (ASQ), a clinical measure of SH, completed a CRT task in the alternating presence and absence of tactile stimulation.

**Results.** Tactile stimulation interfered with attention (i.e., produced longer reaction times), and this effect was significantly greater in participants with more SH (higher ASQ scores). Accuracy on the CRT was high in blocks with and without stimulation. Habituation within stimulation blocks was not detected.

**Conclusion.** This approach can detect distraction from a cognitive task by a steady, faint tactile stimulus that does not degrade response accuracy. The method was also sensitive to the hypothesized enhancement of this effect by SH. These results support the potential utility of this approach to quantifying SH, and suggest possible refinements for future studies.

## INTRODUCTION

A significant percentage of patients with Tourette syndrome (TS) report hypersensitivity to various sensory stimuli (*Cohen & Leckman, 1992*). Uncomfortable awareness of a shirt's cloth tag touching the nape of the neck is a classic example that illustrates two

Corresponding author
Kevin J. Black, kevin@WUSTL.edu

characteristics of sensory hypersensitivity (SH). First, lack of habituation is a key feature, in that people with SH are often bothered by nearly continuous stimuli that most people ignore soon after onset. Second, the annoying sensations are often provoked by stimuli that are faint in intensity and have little apparent value, yet subjects with TS have normal sensory thresholds (*Belluscio et al., 2011*). These features suggest an alteration in central processing of sensation rather than enhanced peripheral detection (*Belluscio et al., 2011*). SH is not specific to TS, as it also occurs commonly in other developmental disorders and in some healthy subjects. However, the pathophysiology of SH is not well understood despite its effect on quality of life (*Ferrão et al., 2012*; *Hazen et al., 2008*; *Kinnealey, Koenig & Smith, 2011*; *Leckman et al., 2006*; *Mangeot et al., 2001*; *Miguel et al., 2000*; *Prado et al., 2008*). Additionally, the common occurrence in TS of SH, premonitory sensations, Attention-Deficit/Hyperactivity Disorder (ADHD), and Obsessive-Compulsive Disorder (OCD) suggest that SH may provide a clue to the pathophysiology of these disorders (*Cohen & Leckman, 1992*; *Ferrão et al., 2012*; *Mangeot et al., 2001*; *Miguel et al., 2000*; *Sutherland Owens, Miguel & Swerdlow, 2011*).

Further progress would be facilitated by the development of objective, quantitative tests of SH. One approach would be to quantify the severity and time course of habituation to an externally applied stimulus. However, direct inquiries about whether one still notices a stimulus draw one's attention to it, thus influencing the measurements in a manner metaphorically similar to the phenomenon described by Heisenberg's uncertainty principle for elementary particles (*Heisenberg, 1927*). The present study tests the hypothesis that SH can be quantified indirectly by measuring and timing the distraction that it exerts during a cognitive task. (We refer to this strategy as "the Ariana effect" for reasons noted in Acknowledgments.)

## METHODS

We recruited a convenience sample of 19 ambulatory adult volunteers, 9 women and 10 men. The age range was 25–63, and the average age of the participants was 35. The study was approved by the Washington University Human Research Protection Office (IRB approval # 201108081), and each subject gave verbal informed consent prior to participation.

To measure attention, participants completed a choice reaction time (CRT) task, consisting of 7 blocks of 30 trials with a mean inter-trial interval of 2.0 s, so that each block lasted approximately one minute. Stimuli were presented on a Dell Latitude D620 14.1″ laptop screen. Each trial consisted of the presentation of either the letter "X" or "O" until the participant responded. Participants were asked to press a corresponding button as quickly and accurately as possible. The keyboard button "1" was labeled with a sticker that had the letter "X" on it, and the keyboard button "2" was labeled with the letter "O". Before the beginning of the task, they were warned that tactile stimulation would be applied during parts of the test, but were not given a verbal warning prior to each application of the stimulation. Between blocks, the phrase "Please wait" was presented briefly on the screen to signal the experimenter to apply or remove the stimulus.

A stimulus was chosen based on the goal of a mildly annoying tactile stimulus that was similar to those people experience daily but that could be applied reproducibly. The tactile stimulation was applied by the researcher with a 1.65 mm diameter nylon monofilament line to the left ankle of the participants during the second, fourth and sixth block of the 7-block CRT task, whereas no stimulation at all was applied during the other blocks. The idea for the nylon monofilament was suggested by von Frey hairs, modified by the intent to eventually adapt the task for administration during functional MRI. The ankle was chosen as an area that would be easily accessible during possible future functional MRI studies. The researcher applying the stimulation sat as far behind the participant as possible, and modified his or her position in relation to the participant as little as possible between stimulation and control blocks.

After administration of the CRT task, participants completed the Adult Sensory Questionnaire (ASQ), a 26-item self-administered questionnaire developed to screen for sensory defensiveness in adults (*Kinnealey, Oliver & Wilbarger, 1995*), with additional questions about age, sex, and history of TS or other tic disorders, ADHD, and OCD.

Data from the first minute of the task was excluded from statistical analysis, as mean reaction times improved rapidly during the first block. (That decision was made after collecting the data but before examining the effect of SH on the change in reaction time.) We measured the interference caused by tactile stimulation by subtracting the average of the median reaction times of each block during the non-stimulation blocks from the average of the median reaction times of each block during the stimulation blocks, and tested for statistical significance using a one-sample $t$ test. An independent-samples $t$ test was used to compare this interference effect in participants with a total ASQ score of $\geq 6$ (moderate or definite sensory defensiveness) to those with a score of $<6$ (no sensory defensiveness) (*Kinnealey, Oliver & Wilbarger, 1995*). We also examined using the Pearson correlation test whether there was a correlation between the slowdown of reaction time during the tactile stimulation blocks and the ASQ score (used as a continuous variable). The accuracy rate was calculated as the number of correct responses divided by the total number of responses, and a one-sample $t$ test was used to test whether accuracy differed significantly between blocks with and without stimulation. We examined whether there was evidence for habituation during the tactile stimulation blocks by comparing the average of the median reaction times of the 10 last trials of each stimulation block to the average of the median reaction time of the first 10 trials for each such block, using a one-sample $t$ test.

## RESULTS

In our sample, 26% (5) of the participants had no sensory defensiveness (ASQ score $<6$), 47% (9) had moderate sensory defensiveness (a score between 6 and 10), and 26% (5) had definite sensory defensiveness (score $>10$). The group with moderate or definite SH included 7 men and 7 women (mean age 36.1 years, SD = 12.84), whereas 3 men and 2 women had no SH (mean age 31.8 years, SD = 9.06). By self-report, 16% (3) had ADHD,

5% (1) had OCD, 11% (2) had TS, 5% (1) had another tic disorder, and 11% (2) had atopic dermatitis.

Across all subjects, the average interference caused by tactile stimulation was 14.3 ms ($M = 14.3$ ms, $SD = 26.31$), i.e., reaction time in the stimulation blocks ($M = 470.2$ ms, $SD = 87.4$) was longer than in the stimulation-free blocks ($M = 455.9$, $SD = 75.3$), $t(18) = 2.37$, $p = .03$ (Fig. 1). The interference effect in participants with moderate or definite SH ($n = 14$; $M = 21.86$; $SD = 25.94$) differed significantly from those with no SH ($n = 5$, $M = -6.80$, $SD = 13.14$), $t(17) = -2.33$, $p = .03$ (Fig. 2). There was a trend towards a significant correlation between the interference effect and the ASQ score ($r = 0.44$, $p = .06$), such that individuals with higher ASQ scores (more hypersensitivity) demonstrated greater interference (Fig. 3).

The average accuracy rate was 98.66% ($M = 98.66\%$, $SD = 1.22\%$) for the stimulation blocks and 98.84% ($M = 98.84\%$, $SD = 1.44\%$) for the stimulation-free blocks, giving an average difference of $-.17\%$ ($M = -.17\%$, $SD = 1.58\%$), which did not differ significantly from zero, $t(18) = -.48$, $p = .64$. Testing for evidence of habituation within each stimulation block, the reaction time difference between the first 10 trials ($M = 473.58$, $SD = 85.68$) and the last 10 trials ($M = 464.76$, $SD = 88.57$) was not significant, $t(18) = 1.29$, $p = .22$.

## DISCUSSION

This pilot study demonstrates the mildly distracting effects of steady tactile stimulation during an attentional cognitive task. The effect of tactile stimulation on reaction times was small but statistically significant. Importantly, people with a higher degree of SH experienced more interference from tactile stimulation than did people with a lower degree of SH ($p = .03$ by $t$ test, $p = .06$ by correlation). Accuracy was nearly perfect with or without tactile stimulation, with no statistically significant difference. We attempted to measure habituation within stimulation blocks; although on average our participants improved their reaction times slightly from the beginning to the end of the stimulation blocks, this improvement was not statistically significant. Overall, our preliminary data provide initial support for the Ariana effect, as people who reported SH experienced a greater decrement in speed performing the task at hand when presented with a faint tactile stimulus.

Our study has several limitations. The administration of the tactile stimulus was not standardized given that the monofilament was applied to the skin by hand, presumably with some variation in the amount of pressure applied. Additionally, the computerized task was not performed in a perfectly controlled testing environment (e.g., the distance from the participant to the computer screen was not controlled, the luminance of the testing room varied, response hand was not controlled). It is possible that habituation to faint tactile stimuli happens so quickly or so slowly that our fixed experimental design (approximately 30 trials per minute) and statistical analysis approach were not able to capture its effects in our small sample. All these limitations, along with our modest sample size, would tend to reduce our power to find significant effects. Despite these limitations,

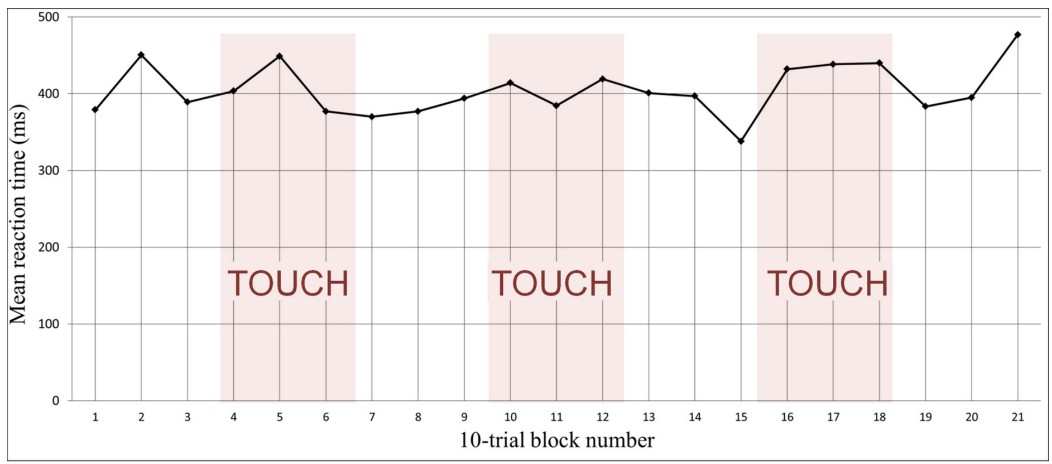

**Figure 1** **Reaction time slows down with tactile stimulation.** Subjects were slower during blocks when the tactile stimulus was applied (identified in the figure by "TOUCH" and a colored background). The line graph shows for each group of 10 trials the mean over subjects of the median reaction time for that group of trials within each subject.

we were able to detect a significant interference effect that was larger in participants with higher SH, consistent with our hypotheses. The results suggest that future studies accounting for these limitations may find even more robust effects.

In interpreting our results the following additional factors should be considered. First, some of the participants reported at recruitment that they had SH, potentially unblinding the researcher applying the tactile stimulation to the degree of SH of these participants. However, although some of these subjects who reported having SH did have a high ASQ score, others did not, and since the ASQ was administered after the cognitive-sensory task, during the task neither the researcher nor the participant was aware of the subject's actual ASQ score. Of course, one could argue that completing the sensory-cognitive task may have influenced the responses of the participants to the ASQ. Second, it is difficult, if not impossible, to completely disentangle sensory expectations from sensory experiences, since people with a high degree of SH probably develop expectations regarding the effects of stimuli through experience. Third, the ASQ was designed as a screening tool for diagnosis in conjunction with a more thorough interview (*Kinnealey, Oliver & Wilbarger, 1995*), so it could be argued that using the ASQ score as a continuous variable for the correlation analysis was not in accordance with its intended design. Finally, one could question how faithfully the monofilament stimulus we chose reflects the sensitivity to daily tactile stimuli such as those assessed in the ASQ. Nevertheless, none of these factors substantially complicate interpreting this pilot study.

In conclusion, this study provides intriguing initial support for the idea that the cognitive distraction provided by faint tactile stimulation may be exploited to quantify SH. Further research is warranted to explore this effect and its relationship with age, sex, and diagnosis. This method may prove useful in studying the role of SH in the pathophysiology and clinical characteristics of Tourette syndrome and other neuropsychiatric disorders.

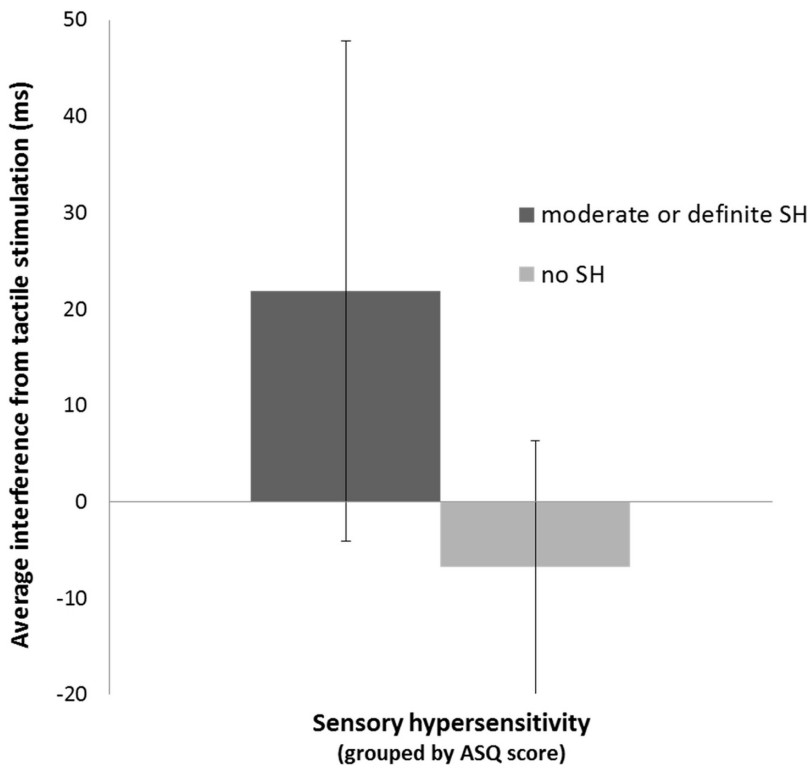

**Figure 2 Interference from tactile stimulation in subjects with *vs.* without SH.** The slowing of reaction time during tactile stimulation ("interference") was greater in subjects with sensory hypersensitivity as defined by the Adult Sensory Questionnaire.

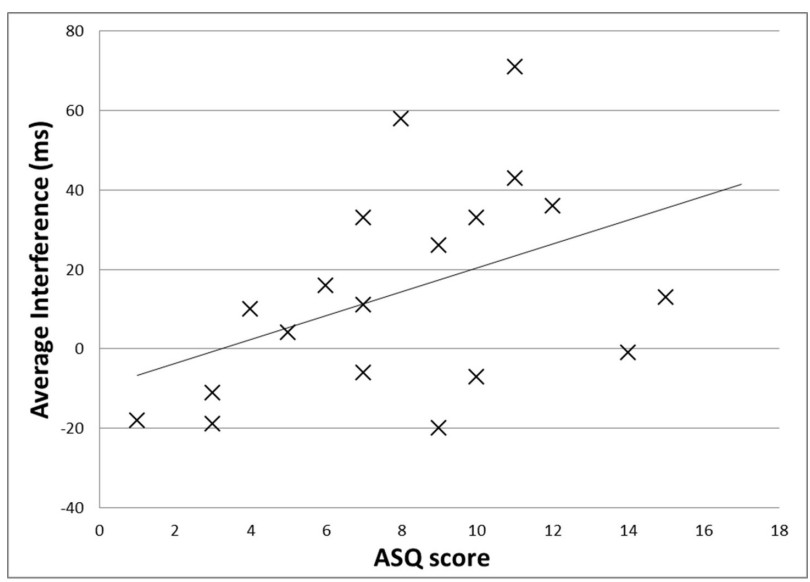

**Figure 3 Interference from tactile stimulation, by ASQ score.** The slowing of reaction time during tactile stimulation ("interference") tended to be greater in subjects with higher scores on the Adult Sensory Questionnaire.

## ACKNOWLEDGEMENTS

The authors thank Ariana Jane Black, who proposed this experimental approach to quantifying SH after observing that SH is problematic in real life largely because it distracts one from cognitive tasks such as school examinations. Dr. Moya Kinnealey kindly provided the ASQ and related information.

### Funding
This study was funded in part by the U.S. National Institutes of Health (K24 MH087913). The funders had no role in study design, data collection and analysis, decision to publish, or preparation of the manuscript.

### Grant Disclosures
The following grant information was disclosed by the authors:
U.S. National Institutes of Health: K24 MH087913.

### Competing Interests
Dr. Black is an Academic Editor for PeerJ.

### Author Contributions
- Vassilis N. Panagopoulos conceived and designed the experiments, performed the experiments, analyzed the data, wrote the paper.
- Deanna J. Greene conceived and designed the experiments, performed the experiments, analyzed the data, contributed reagents/materials/analysis tools, reviewed and edited the manuscript.
- Meghan C. Campbell conceived and designed the experiments, analyzed the data, contributed reagents/materials/analysis tools, reviewed and edited the manuscript.
- Kevin J. Black conceived and designed the experiments, performed the experiments, analyzed the data, reviewed and edited the manuscript.

### Human Ethics
The following information was supplied relating to ethical approvals (i.e., approving body and any reference numbers):

Washington University Human Research Protection Office, approval # 201108081.

### Supplemental Information
Supplemental information for this article can be found online at http://dx.doi.org/10.7717/peerj.121.

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
