# Peer review of "Towards objectively quantifying sensory hypersensitivity: a pilot study of the “Ariana effect”"

_PeerJ, doi:10.7717/peerj.121_

## Round 0.1 · original submission · Major Revisions

I am concerned that “The researcher applying the tactile stimulation was not always blind to the degree of SH of the participants.” This may have created unconscious bias in hand-delivered tactile stimulation. I think that what you should do is to run additional subjects with research assistants that are completely blind to the degree of SH and even to the rationale of the study. You should also probably remove from data analysis the subjects that have been stimulated by researchers not blind to the SH of the participants. Having a larger number of subjects tested in each group and by research assistants blind to SH level of participants should address some of the concerns of the reviewers.

You should also carefully address the excellent constructive criticisms of the reviewers, including the ones on statistical analyses and potentially distracting effects of movements from the person delivering the tactile stimulation.

·

Basic reporting

Please check the Title of the figures for clarity

Experimental design

No comments

Validity of the findings

No Comments

·

Basic reporting

Abstract:

Point 1:

Authors states: “In an attempt to objectively quantify SH severity, the authors tested whether a steady tactile stimulus distracts subjects with SH from a cognitive task more than it does subjects without SH.”

It seems obvious that patients with SH should be more distracted. It’s like asking if people with a snake phobia will be more distracted by a snake on their desk than people without a snake phobia. Perhaps, it makes more sense to state something to the effect of:

“In an attempt to objectively quantify SH severity, the authors tested the extent to which steady tactile stimulus distracts subjects with SH versus subjects without SH during the performance of choice response task.”

The key contribution here seems to be to demonstrate a developing method to quantify SH severity. I don’t know this literature, but it seems obvious that a tactile stimulus should distract an SH sample more so than controls.

Introduction:

Point 1:
It is not clear whether the main objective is to test the hypothesis that tactile stimulation will distract SH patients more than controls, or to assess a potential method for quantifying SH severity that may have clinical and investigational utility.

Perhaps, the authors could clarify that a positive relationship between SH severity and distractibility via tactile stimulation is expected, and that the primary goal of this work is to assess whether that dynamic can be behaviorally quantified.


Figures

Figure 1
The authors’ state: “The line graph shows for each group of 10 trials the mean over subjects of the median reaction time for that block within each subject.”

I am very confused by this image. Blocks were 30 trials long. RT scores were pulled from the median within those blocks and then averaged per condition. This figure is not showing trail-based data, nor is it showing block based data, but seems to be showing RT data for successive groupings of 10 trials? I don’t see how this is relevant to the reported results. Perhaps, I am missing something here?

Experimental design

Method:

Point 1
It would be nice to know the inter-trial interval.

Point 2
The interpretation of results requires considering potential attentional confounds associated with sensory expectation versus active SH effects alone (stimulations condition was not randomized). Perhaps this should be mentioned in the limitations.

Point 3
Why was the ASQ administered after the CRT task? The task-taking experience may have biased the ratings. This should perhaps be addressed in the limitations.

Point 4
Seems that block one served as a training block. Why not just call it that?

Point 5
I am very confused as to why the authors used median RT per block to create their averaged RT per condition. More typically, one would average across trials per condition. In any case, I don’t see this as wrong per se, but it is unusual, and as such, the authors may want to provide some basis for choosing that approach.

Point 6
Regarding the assessment of group differences on stimulation effects (i.e., the difference score), it seems a repeated measures ANOVA would be suitable (group x condition, with RT as the dependent). That being said, the methods seem ok, just unusual. Still, given the total sample is only 19, the authors may want to report directly in the methods what the resultant n-sizes per group are after making this split. Also, some information here about how the threshold was chosen may be helpful (i.e., based on clinical diagnostic cut-offs?). It would also be useful to note the gender ratios and ages in the resultant groups. I imagine they are different, and it would be nice to be convinced that these are not driving the results.

Point 7
Because the sample size is small, it seems optimal to assess the relationship between stimulation-distraction and SH rating using the ASQ as a continuous measure. This was done with the correlation analysis. However, this doesn’t tell us if the correlation is driven by one condition in particular (i.e., hopefully by the stim condition). It would be nice (not necessary) to see separate correlation analyses between ASQ and each condition. Alternatively, one might consider a regression looking at the interaction of block-type x ASQ predicting RTs.

Point 8
Why was median RT collected within blocks (and then averaged), but accuracy assessed across blocks?

Point 9
In the analysis of within block effects, again, why use median versus mean? If there are large effects loaded in particular aspects of a block (e.g., last few and/or first few trials), it seems median scores would reduce the chance of finding such effects.

Validity of the findings

Results

Point 1
With the small n size in the no-SH group, I’m wondering how the analyses would have turned out lumping the ‘fence sitters’ into a ‘low-SH’ group (no-SH+moderate-SH vs. clinical-SH). In any case, given the very low sample size of the no-SH group, the continuous approach makes the most sense (i.e., used for the correlation analysis, or the regression I suggested). Also, the authors might consider reporting the mean difference score for each of the 3 diagnostic groups (no SH, marginal SH, and definite SH) so the readers can see the pattern themselves. Finally, if there’s any reason to suspect gender and age differences might be impacting the group analysis, this should be addressed.


Discussion

Point 1
The authors state, “All these limitations, along with our modest sample size, would not bias the results, but would tend to reduce our power to find significant effects.”

I disagree. With only 5 subjects in the no-SH group, variability in the delivery of stimulation could have biased results, as well as, other factors that were not controlled (e.g., response hand relative to side of stimulation, gender, age, etc.).

Point 2
The discussion seems to suggest that evaluating whether SH subjects would show a larger interference effect is the main objective. Perhaps it is, but this should be clarified in relation to the core objective of developing a means to quantify SH severity.

In my view, the methods development narrative is more compelling. With only 5 subjects in the no SH group, and the correlation being weakly significant, the relevance seems more about the development of a new approach to quantify SH severity.

Reviewer 3 ·

Basic reporting

This is a pilot study testing the hypothesis that somatic hypersensitivity influences the performance on cognitive tasks requiring sustained attention. The idea behind the study rationale is logical and interesting. The study is sufficiently well written and clear in all its parts. Results are potentially interesting, although I have some reservations on the study design.

Experimental design

My major concern is with the way the distracting tactile stimuli were administered. If these were delivered by hand by one of the researchers, how were participants blinded to movements performed by the person who delivered the stimuli? In other words, how did the authors make sure that distraction was uniquely dependent on the tactile stimuli and not also by the close presence of the person delivering the stimuli? More details on the experimental setting are necessary.

Validity of the findings

Another major concern comes from the heterogeneous clinical sample. Some patients have ADHD, others OCD, others TS, others apparently none of these neurodevelopmental disorders. There was no attempt to adjust for potential confounding of primary diagnosis (or, even better, of the degree of inattention) on the performance on a reaction time task that is structured like a continuous performance task, thus potentially impaired at baseline in patients with attention deficit.

Additional comments

Hand-delivered stimuli represent a great source of variability, and definitely less than ideal for a sound paradigm. In improving the paradigm for future studies, the authors should avoid this limitation.

---

## Round 0.2 · accepted · Accept

Thanks for addressing all our concerns and for submitting your work to PeerJ.

·

Basic reporting

nothing

Experimental design

nothing

Validity of the findings

nothing

Additional comments

nothing